# Dimensionless Analysis of the Spatial–Temporal Coupling Characteristics of the Surrounding Rock Temperature Field in High Geothermal Roadway Realized by Gauss–Newton Iteration Method

**Jiale Zhou** [1], **Yuan Zhang** [1,2,*], **Peng Shi** [1] and **Yang Liu** [1]

1   School of Mines, China University of Mining and Technology, Xuzhou 221116, China; ts21020076a31tm@cumt.edu.cn (J.Z.); tb22020024a41@cumt.edu.cn (P.S.); 01210291@cumt.edu.cn (Y.L.)
2   State Key Laboratory of Coal Accurate Exploration and Intelligent Mining, China University of Mining and Technology, Xuzhou 221116, China
*   Correspondence: zhangyuan@cumt.edu.cn; Tel.: +86-135-8539-1064

**Featured Application: This study is of great significance for identifying and understanding the spatio-temporal evolution of the temperature field of high geothermal surrounding rock and for effective risk and disaster management in deep mine exploration.**

**Abstract:** Understanding the time–space coupling characteristics of the surrounding rock temperature field in high geothermal roadways is essential for controlling heat damage in mines. However, current research primarily focuses on individually analyzing the temperature changes in the surrounding rock of roadways, either over time or space. Therefore, the Gauss–Newton iteration method is employed to model the coupling relationship between temperature, time, and space. The results demonstrate that the dual coupling function describing the temperature field of the surrounding rock in both time and space provides a more comprehensive characterization of the temperature variations. Over time, as ventilation duration increases, the fitting degree of the characteristic curve steadily rises, and the characteristic curve descends overall. In the spatial dimension, the fitting degree of the characteristic curve gradually decreases with the rise of the dimensionless radius, and the characteristic curve ascends overall. Additionally, as thermal conductivity increases, the fitting degree of the characteristic curve steadily rises.

**Keywords:** high geothermal; surrounding rock temperature field; space–time coupling; dimensionless; Gauss–Newton iteration

## 1. Introduction

Coal, as an indispensable fossil fuel for social development, has long been heavily relied upon worldwide, leading to the depletion of surface resources and the shift of coal mining to deeper levels [1,2]. Yet, as mining depth increases, the virgin rock temperature (VRT) gradually rises, leading to an increasingly prominent issue of mine heat damage [3,4]. Studies have indicated that prolonged exposure to high-temperature and high-humidity environments can result in a decline in human physical fitness, making individuals susceptible to heat-related illnesses such as heat stroke, heat halo, central nervous system disorders, and even death. This significantly impacts miners' physical and mental health and jeopardizes the safe and efficient production of mines [5,6].

The essence of controlling heat damage in a high-temperature mine involves addressing three fundamental questions: Why does heat occur? How high is the temperature? How can heat damage be controlled? The primary focus of research on the first issue centers around analyzing the geothermal geological system of the mine. For instance, Peng et al. [7] conducted surface borehole temperature measurements and groundwater

temperature measurements in the Chenghe mining area of the Weibei coalfield. They combined these findings with thermal conductivity tests on coal rock. The study revealed that, when influenced by critical factors like fractures, folds, the thermal conductivity of coal rock, and groundwater activity, the geothermal gradient and geothermal flow value exhibit an increasing trend from northwest to southeast. Yin et al. [8], based on 14 sets of water samples from the eastern section of the copper well mining area of Yinan Gold Mine, utilized the mineral saturation index and geochemical geothermal temperature scale to analyze thermal storage temperature and recharge characteristics. This aimed to unveil the study area's geothermal water fugitive transmission pattern and genesis mechanism and propose a geothermal water genesis model. The inference is that the recharge area consists of precipitation in the Northwest Mountain range and the surrounding atmosphere. The observation is that it receives heat from the surrounding rocks. The water level rises at the fracture zone intersection and is stored in the lower and middle Cambrian thermal reservoirs, continuing to receive heat from deeper heat sources. In the Qingdong coal mine in China, Xu et al. [9] gathered 13 samples of geothermal water (>34 °C) and 11 samples of ordinary groundwater (>0 °C). The findings reveal that atmospheric precipitation and water–rock interaction are the region's primary sources of geothermal water. Addressing the second question involves elucidating the cooling load in the high-temperature areas of the mine and establishing the foundation for devising a cooling scheme [10,11]. Hence, it is crucial to investigate the temporal and spatial evolution characteristics of the surrounding rock temperature field in high geothermal roadways influenced by various factors. Over the past two decades, research on the temporal and spatial evolution characteristics of the temperature field in the surrounding rock of high geothermal roadways has been growing, yielding fruitful results. Zhu et al. [12] designed a series of physical similarity simulation experiment equipment for studying the heat conduction and heat transfer of surrounding rock in a mine roadway. The equipment is used to systematically and quantitatively analyze the influence of physical parameters on the temperature field of the surrounding rock and its dynamic changes. It was found that when the temperature field tends to be stable, the airflow velocity is closely related to the temperature distribution of the surrounding rock. When the temperature curve tends to be stable, the thermal diffusivity of the rock also has a great influence on the temperature curve but has little effect on the shape of the curve. Dong et al. [13] employed an independently developed heterogeneous surrounding rock test simulation system to analyze the impact of periodic wind temperature on the evolution of the surrounding rock temperature field. The results indicate that, under periodic ventilation conditions, the relationship between the surrounding rock temperature and time aligns with the cosine function relationship. Zhu et al. [14] explored the impact of seasonal wind temperature on the evolution patterns of the temperature field in the surrounding rock of roadways. The results show that the surrounding rock can be adjusted due to the influence of seasonal temperature differences. In their study, Qin et al. [15] formulated a mathematical model for the unsteady temperature field of roadway-surrounding rock grounded in energy conservation principles and Fourier's law. They developed a solution program and utilized the finite volume calculation method to solve the temperature field of the roadway-surrounding rock. Tian et al. [16] examined the correlation between the temperature field of the roadway-surrounding rock and the depth and ventilation time of the surrounding rock. They noted that the farther away from the roadway wall, the more stable the temperature change. The shape and size of the heat-regulating ring visibly change over time. Chen et al. [17] conducted experiments on the thermal characteristics of surrounding rock based on a mathematical model of surrounding rock heat transfer in deep underground spaces. They examined the effects of heat transfer time, air temperature, and radial displacement on temperature distribution. In their study, Wang et al. [18] developed a mathematical model for deep roadway-surrounding rock. They investigated the impact of the insulation layer on the temperature field inside and outside the tunnel. The results indicated that the disturbance range of wall temperature and surrounding rock temperature decreased due to the insulation layer. When dealing with data obtained from experiments

or numerical simulations, many researchers analyze correlations among the data through fitting. For instance, using a Yunnan roadway as a case study, Xu et al. [19] observed that the temperature field of the roadway-surrounding rock with secondary lining exhibited a rapid increase, rapid decrease, stability, and slight decrease over time. In the spatial dimension, the temperature in the roadway follows an 'n' shape along the longitudinal direction, and the temperature profile of the surrounding rock is parabolic. Zhang et al. [20,21] examined the spatio-temporal evolution pattern of the dimensionless surrounding rock temperature field using physical modeling tests and performed nonlinear fitting of the data using some mathematical software. They observed that the dimensionless temperature and dimensionless radius of the roadway-surrounding rock exhibited an exponential function relationship, while the dimensionless temperature and dimensionless time approximated the Hill function relationship. Zhang et al. [22] formulated a numerical model for the temperature field of roadway-surrounding rock employing the finite difference method. Nonlinear fitting of the numerical simulation results revealed the relationship between the disturbance range of the surrounding rock temperature field and ventilation time. According to the case analysis, when the disturbance range is $R = 13.6$, the temperature disturbance range does not significantly expand after four years of ventilation. As the above shows, the fitting relationship between surrounding rock temperature and time or space is typically nonlinear. In addressing nonlinear fitting problems, many researchers commonly employ the Gauss–Newton method or its enhancements. For instance, Jamroen et al. [23] introduced an effective curve-fitting technique based on the Gauss–Newton (GN) method with a variable damping coefficient to assess the voltage parameters of the lightning pulse. This method can be applied to derive the fundamental curve of the complete lightning pulse voltage, corrected for oscillation, from the test case parameters in accordance with the standard (IEC 61083-2) [24]. To determine the characteristic parameters of moving magnetic targets using existing data, Ge et al. [25] introduced a real-time detection method for distributed scalar sensor networks. This method employs a hybrid algorithm that combines particle swarm optimization (PSO) with the Gauss–Newton method. This method can utilize the Gauss–Newton method to obtain more precise solutions using the high-precision advantage of particle swarm optimization.

The above research mainly analyzes the temperature change in roadway-surrounding rock on a single scale of time or space. While this accurately describes the evolution pattern of the surrounding rock temperature field, it lacks universality. Thus, it is essential to investigate the coupling relationship between the temperature field of the roadway-surrounding rock and both time and space. However, few reports exist on such research. Building on this, the paper establishes a heat conduction model for roadway-surrounding rock, simulates the temporal and spatial evolution patterns of the temperature field, and compares these with previous studies to validate the reliability of the model proposed in this paper. Furthermore, the heat conduction equation and single-value condition of the temperature field are dimensionless. Ultimately, the Gauss–Newton iteration method is employed to establish the coupling relationship between temperature and both time and space.

## 2. Analysis Method of Time–Space Coupling Characteristics of Surrounding Rock Temperature Field in High Geothermal Roadway

### 2.1. Modeling Method

The primary focus of this paper is the study of the general heat transfer process of roadway-surrounding rock. Hence, the impact of the wind flow field on the surrounding rock temperature is disregarded. Additionally, the physical properties of the surrounding rock are assumed to be isotropic, and equivalent values are assigned to each parameter. Furthermore, the following assumptions are made:

(1) The heat transfer inside the surrounding rock is only in the form of heat conduction, and there is no other internal heat source.

(2) We ignore the influence of groundwater on the temperature field of the surrounding rock.

(3)  The initial temperature of the roadway-surrounding rock is considered the VRT.

With the COMSOL 6.0 numerical simulation software, a two-dimensional numerical model of the surrounding rock of a high geothermal roadway is established to simulate the evolution of the surrounding rock temperature with time and space after 361 days of ventilation. As shown in Figure 1, the model features a 50 m width and a 40 m height. The roadway is rectangular and located in the center of the model, with a width of 5 m, a height of 4 m, and a hydraulic radius of 2.2 m. The density of the roadway-surrounding rock is set to 3000 kg/m$^3$, with a specific heat capacity of 500 J/(kg·K).

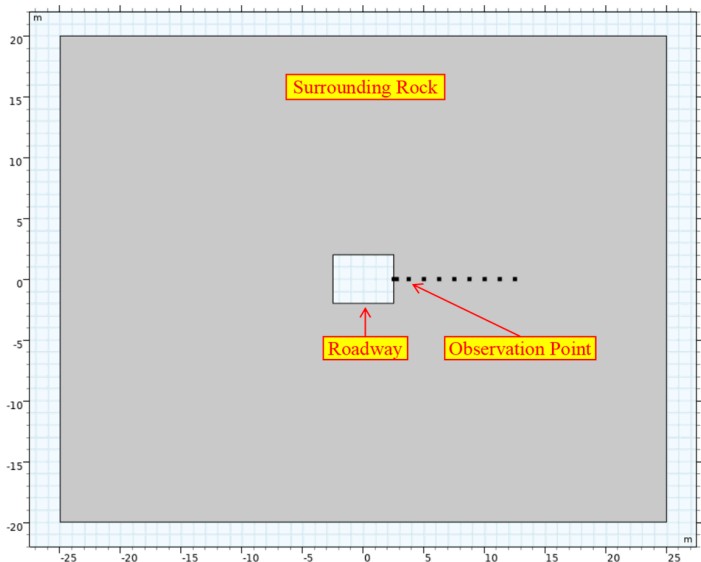

**Figure 1.** Temperature field two-dimensional model of surrounding rock of roadway with high geothermal.

Nine observation points, spaced at intervals of 0.5 times the hydraulic radius, are established. The temperature variations in the surrounding rock at these points are recorded at a position halfway up the roadway, extending horizontally from the right side of the rock wall into the deeper region of the surrounding rock. (Because the dimensionless radius $R = 0$ at the rock wall surface cannot be brought into the following binary coupling function for fitting and solving, an additional point is specially added at 0.1 times the hydraulic radius near the wall surface, for a total of 10 observation points.) The virgin rock temperature ($T_0$) is set at 50 °C, and the model's outer boundary is a fixed temperature boundary. Forced convection heat transfer occurs between the roadway walls. The wind speed is set at 2 m/s at a temperature of 25 °C.

### 2.2. Simulation Scheme

In order to explore the influence of thermal conductivity on the temperature field of roadway-surrounding rock, three simulation programs with different thermal conductivity of surrounding rock were designed. Considering that $a = \lambda/\rho c$, the thermal diffusivity coefficients of each scheme also vary ($a$ is the thermal diffusivity coefficient of the roadway-surrounding rock in m$^2$/s, $\lambda$ is the thermal conductivity of the roadway-surrounding rock in W/(m·K), $c$ is the specific heat of the roadway-surrounding rock in J/(kg·K)).

As suggested in the literature [22], the value of $h$ can be calculated using the following equation:

$$Bi = 0.0115\frac{\lambda_f}{\lambda}(av)^{-0.4}(ud)^{0.8} \tag{1}$$

$$Bi = \frac{hr_0}{\lambda} \tag{2}$$

where $Bi$ is the Biot number, $\lambda_f$ is the thermal conductivity of the airflow at 0.026 W/(m·K), $\lambda$ is the thermal conductivity of the roadway-surrounding rock in W/(m·K), $a$ is the thermal

diffusivity coefficient of the roadway-surrounding rock in m$^2$/s, $\nu$ is the kinematic viscosity coefficient of the air flow at $14.4 \times 10^{-6}$ m$^2$/s, $u$ is the wind speed in m/s, and $d$ is the hydraulic diameter in m.

The unstable heat transfer coefficient can be obtained by connecting Equations (1) and (2). The specific parameters are shown in Table 1.

**Table 1.** Simulation scheme.

| Program | Thermal Conductivity of the Surrounding Rock | Thermal Diffusivity Coefficient of the Surrounding Rock | Unstable Heat Transfer Coefficient |
|---------|----------------------------------------------|---------------------------------------------------------|------------------------------------|
| 1 | $\lambda = 1$ W/(m·K) | $a = 0.67 \times 10^{-6}$ m$^2$/s | $h = 26.91$ W/(m$^2$·K) |
| 2 | $\lambda = 2$ W/(m·K) | $a = 1.33 \times 10^{-6}$ m$^2$/s | $h = 15.00$ W/(m$^2$·K) |
| 3 | $\lambda = 3$ W/(m·K) | $a = 2.00 \times 10^{-6}$ m$^2$/s | $h = 12.74$ W/(m$^2$·K) |

*2.3. Nondimensionalization of Single-Value Condition of Temperature Field*

To elucidate the general patterns of temporal and spatial evolution in the temperature field of roadway-surrounding rock, reference [21] non-dimensionally characterized the thermal conductivity equation of the roadway-surrounding rock temperature field and its univariate conditions. The specifics are outlined below:

$$\begin{cases} \Theta = \left(T - T_f\right)/\left(T_0 - T_f\right) \\ R = (r - r_0)/r_0 \\ Fo = a\tau/r_0{}^2 \end{cases} \tag{3}$$

where $\Theta$ is the dimensionless temperature, $R$ is a dimensionless radius, $T$ is the temperature of the roadway-surrounding rock in °C, $T_f$ is the airflow temperature in °C, $T_0$ is the VRT of the roadway in °C, $r$ is the distance between the measuring point and the center of the roadway along the horizontal direction in m, $r_0$ is the hydraulic radius of the roadway in m, $Fo$ is dimensionless time, $a = \lambda/\rho c$ is the thermal diffusivity coefficient of roadway-surrounding rock in m$^2$/s, and $\tau$ is the ventilation time in s.

*2.4. Time–Space Coupling Method of Temperature Field Based on Gauss–Newton Method*

The temporal and spatial evolution law of the temperature field of the surrounding rock of the roadway can be regarded as a binary function of time ($Fo$) and space ($R$), that is, $\Theta = f(Fo, R)$. This function needs to fit the unquantified surrounding rock temperature value in time and space. This paper uses the Gauss–Newton method to fit [26,27]. The principle is as follows:

Suppose there are $m$ known temperature data, which are $\Theta_1, \Theta_2, \ldots, \Theta_m$. There are $n$ parameters to be solved, which are $a_1, a_2, \ldots, a_n$, and then the estimated output for each set of independent variables is $\hat{\Theta}_1, \hat{\Theta}_2, \ldots, \hat{\Theta}_m$. The solution of the time–space coupling characteristics of the temperature field is a process of gradually approaching the estimated output value to the actual output value, that is, $\hat{\Theta}_i - \Theta_i \to 0$. Therefore, the solution to the above problem can be transformed into a multidimensional optimization problem:

$$\min_{\hat{a}_1,\hat{a}_2,\cdots,\hat{a}_n} \sum_{i=1}^{m}\left(\hat{\Theta}_i - \Theta_i\right)^2 \tag{4}$$

Let $r_i = \hat{\Theta}_i - \Theta_i$, then Formula (2) can be transformed into

$$\min \sum_{i=1}^{m}\left(r_i(a_1, a_2, \cdots a_n)\right)^2 \tag{5}$$

Let $r = [r_1 \; r_2 \cdots r_m]^T$, then

$$\min \sum_{i=1}^{m} (r_i(a_1, a_2, \cdots a_n))^2 = r(a)^T r(a) \tag{6}$$

The Gauss–Newton method is used to solve Formula (4) iteratively, then

$$a^{(l+1)} = a^{(l)} - \left(J(a^{(l)})^T J(a^{(l)})\right)^{-1} J(a^{(l)})^T r(a^{(l)}) \tag{7}$$

where $l$ is the number of iterations and $J(a^{(l)})$ is the Jacobi matrix of $r$ at the $l$th iteration.

There is a problem of $J(a)^T J(a)$ non-positive definite matrix in the solution process of Formula (5), so the Levenberg–Marquardt algorithm is used to correct it, as shown in Formula (6):

$$a^{(l+1)} = a^{(l)} - \left(J(a^{(l)})^T J(a^{(l)}) + \mu_l I\right)^{-1} J(a^{(l)})^T r(a^{(l)}), \tag{8}$$

where $\mu_l$ is the damping factor and $I$ is the unit matrix.

References [20,21] pointed out that the relationship between dimensionless temperature and dimensionless radius conforms to the exponential relationship with e as the base, and the relationship between dimensionless temperature and dimensionless time conforms to the Hill equation relationship, as shown below:

$$\Theta = I \times e^{DR} + 1 \tag{9}$$

$$\Theta = \frac{E}{Fo^G + H} + F \tag{10}$$

where D, E, F, G, H, and I are all undetermined parameters. According to the determined values of $R$ and $Fo$, these undetermined parameters will obtain specific values.

With reference to [28], the specific values of the uncertain parameters in Equation (10) under the conditions $\lambda = 1.2$ W/(m·K), $a = 0.8 \times 10^{-6}$ m$^2$/s ($a = \lambda/\rho c$) are obtained, and the results are shown in Table 2.

**Table 2.** The values of undetermined parameters E, G, and H under various $R$ conditions.

| Dimensionless Radius/$R$ | E | G | H |
|:---:|:---:|:---:|:---:|
| 1 | 0.9529 | −0.735 | 1.1236 |
| 2 | 0.1827 | −0.987 | 0.2942 |
| 3 | 0.0491 | −1.2149 | 0.1023 |
| 4 | 0.0147 | −1.4448 | 0.0389 |
| 5 | 0.0045 | −1.681 | 0.0152 |

Plot the relationship curve between the dimensionless radius $R$ and parameters E, G, and H based on the data in Table 2, as illustrated in Figure 2. It is evident that the dimensionless radius $R$ exhibits an exponential function relationship with parameter E, a linear relationship with parameter G, and an exponential function relationship with parameter H.

Figure 2 shows that the relationship between dimensionless temperature, dimensionless time ($Fo$), and dimensionless radius ($R$) is $\Theta = f(Fo, R)$. Therefore, Formulas (7) and (8) can be coupled, that is,

$$\Theta = \frac{-\exp(A \times R^B)}{O \times R \times Fo^P + \exp(S \times R^Q)} + C \tag{11}$$

As the dimensionless radius $R = 0$ at the wall surface of the surrounding rock yields no solution to Equation (11), the applicable range of the coupling formula is $R > 0$. In scenarios where excavation and ventilation are not conducted, the rock maintains its original thermal

balance, and the temperature of the rock everywhere is the VRT. By inserting this value into Formula (11), C = 1 can be obtained as follows:

$$\Theta = \frac{-\exp(A \times R^B)}{O \times R \times Fo^P + \exp(S \times R^Q)} + 1, R > 0 \tag{12}$$

The Gauss–Newton iteration form of Formula (12) is

$$\begin{aligned}
\boldsymbol{a}_{A,B,O,P,Q,S}^{(l+1)} = \ &\boldsymbol{a}_{A,B,O,P,Q,S}^{(l)} \\
&- (\mathbf{J}(\boldsymbol{a}_{A,B,O,P,Q,S}^{(l)})^{\mathrm{T}} \mathbf{J}(\boldsymbol{a}_{A,B,O,P,Q,S}^{(l)}) \\
&+ \mu_l \mathbf{I})^{-1} \mathbf{J}(\boldsymbol{a}_{A,B,O,P,Q,S}^{(l)})^{\mathrm{T}} \boldsymbol{r}(\boldsymbol{a}_{A,B,O,P,Q,S}^{(l)})
\end{aligned} \tag{13}$$

where A, B, O, P, Q and S are the parameters to be solved in the binary function, $l$ is the number of iterations, $\mathbf{J}(\boldsymbol{a}^{(l)})$ is the Jacobi matrix of $\boldsymbol{r}$ at the $l$th iteration, $\mu_l$ is the damping factor, and $\mathbf{I}$ is the unit matrix.

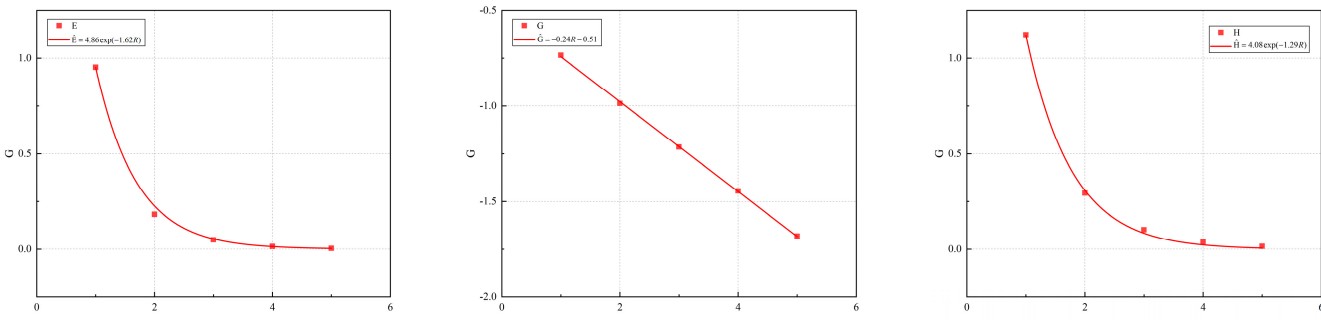

**Figure 2.** The relationship curve between $R$ and parameters E, G, and H.

## 3. Results and Analysis

### 3.1. Temporal and Spatial Evolution of Temperature Field

　　Figure 3 illustrates the relationship curve between the dimensionless temperature and dimensionless radius of the roadway-surrounding rock under the specified conditions: air temperature of 25 °C, VRT of 50 °C, and thermal conductivity of 2 W/(m·K). It is observed that, as the surrounding rock extends radially to the deep, the dimensionless temperature gradually increases, and the rate of temperature rise gradually decreases until it reaches the initial virgin rock temperature. Additionally, the figure indicates that, in the early stages of ventilation, the roadway's surrounding rock is minimally affected by disturbances. However, as the ventilation time extends, the cooling range of the surrounding rock continues to expand. For ventilation periods of 1 day, 31 days, 181 days, and 361 days, the cooling radius of the surrounding rock is <1 $R$, 3 $R$, 6 $R$, and 7 $R$, respectively.

　　Furthermore, as ventilation time extends, the temperature of the roadway wall gradually decreases, but the rate of temperature drop also decreases. As ventilation time progresses from 1 day to 31 days, 181 days, and 361 days, the wall temperature of the roadway is reduced by 45%, 82%, 90%, and 92%, respectively, compared to the virgin temperature. This occurs because, after the ventilation of the roadway, a temperature difference arises between the surrounding rock and the airflow, leading to convective heat transfer. This disrupts the virgin thermal equilibrium state, and the airflow carries away the heat from the surrounding rock, resulting in a decrease in its temperature. With the continuation of ventilation, the temperature difference between the surrounding rock and the roadway wall gradually increases, and the cooling range of the surrounding rock extends to its interior. However, the extension rate gradually decreases until a new equilibrium state is reached.

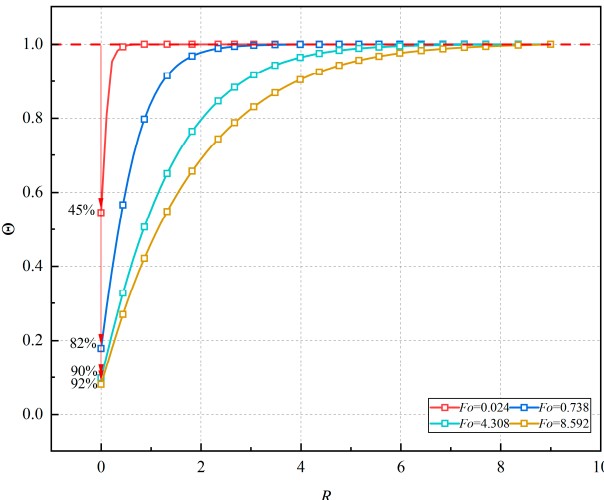

**Figure 3.** The relationship curve between dimensionless temperature and dimensionless radius.

Figure 4 illustrates the relationship curve between dimensionless temperature and dimensionless time of the roadway-surrounding rock under specified conditions: air temperature of 25 °C, VRT of 50 °C, and thermal conductivity of 2 W/(m·K). It is evident that, with an increase in ventilation time, the dimensionless temperature of the surrounding rock gradually decreases, and the cooling rate also gradually decreases, eventually stabilizing at a fixed value. Additionally, the figure shows that the degree of disturbance in the dimensionless temperature of the surrounding rock gradually decreases with an increase in dimensionless radius. After 361 days of ventilation, as the dimensionless radius increased from $R = 0$ to $R = 3$, the temperature of the surrounding rock decreased by 92%, 54%, 31%, and 18%, respectively, compared to the virgin temperature.

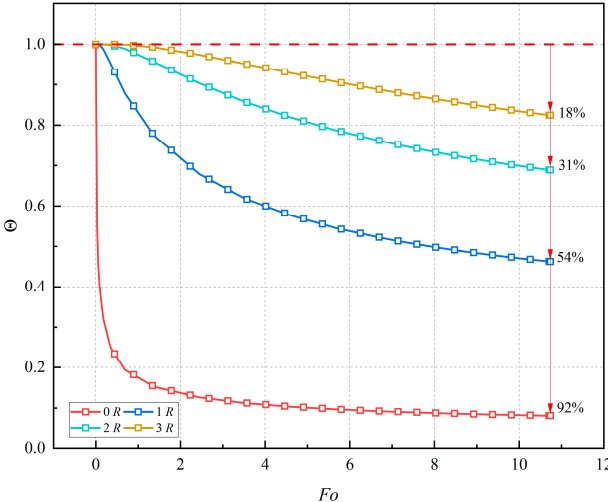

**Figure 4.** The relationship curve between dimensionless temperature and dimensionless time.

### 3.2. The Influence of Thermal Conductivity on the Temperature Field of Surrounding Rock

Figure 5 depicts the spatial distribution curve of the dimensionless temperature of the roadway-surrounding rock under varying thermal conductivity. The change curve of the dimensionless temperature of the surrounding rock with a dimensionless radius is mainly consistent with the observations mentioned above. Simultaneously, as thermal conductivity increases, the dimensionless temperature of the surrounding rock near the roadway wall gradually rises, while the dimensionless temperature of the surrounding rock farther from the roadway wall gradually decreases. As an illustration, considering $R = 0$ and $R = 1$, when ventilated for 31 days with an increase in thermal conductivity

from 1 W/(m·K) to 3 W/(m·K), the dimensionless temperature of the surrounding rock at $R = 0$ decreased by 91%, 90%, and 85%, respectively, compared to the original temperature. Simultaneously, the dimensionless temperature of the surrounding rock at $R = 1$ decreased by 6%, 13%, and 17%, respectively, compared to the VRT. Regarding the observations above, reference [29] provides a clear explanation. With the increase in the thermal conductivity of the surrounding rock, the heat conduction intensity within the surrounding rock gradually increases, as does the convective heat transfer intensity between the roadway wall and the airflow. However, the heat convection intensity surpasses the heat conduction intensity, leading to more significant heat loss from the roadway wall than the heat supply. As a result, the temperature drop rate gradually decreases with the increase in thermal conductivity, while the temperature drop rate inside the surrounding rock increases with the increase in thermal conductivity. Additionally, the increase in thermal conductivity can extend the cooling range of the surrounding rock and postpone the thermal equilibrium time of the surrounding rock temperature field. For instance, considering ventilation for 361 days, with thermal conductivity increasing from 1 W/(m·K) to 3 W/(m·K), the cooling range of the surrounding rock temperature field (temperature drop value > 0.01% of the original temperature range) was 5.5 $R$, 6.5 $R$, and 7 $R$, respectively. This is attributed to higher thermal conductivity leading to faster heat transfer within the surrounding rock, resulting in a more extensive range of disturbed surrounding rock under the same ventilation time.

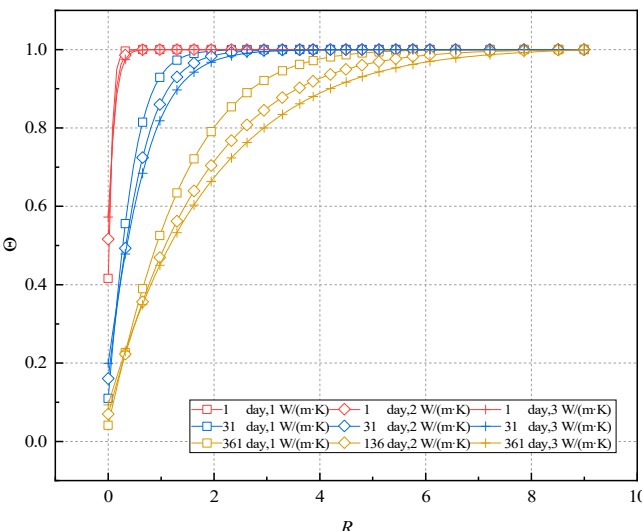

**Figure 5.** The relationship between dimensionless temperature and dimensionless radius under different thermal conductivities.

Figure 6 illustrates the time distribution curve of the dimensionless temperature of the roadway-surrounding rock under varying thermal conductivity. It is evident that, with the increase in dimensionless time under the same thermal conductivity, the dimensionless temperature gradually decreases, and the temperature drop rate also gradually decreases, ultimately reaching a constant value. Simultaneously, with an increase in thermal conductivity, the temperature drop rate of the surrounding rock gradually decreases, and the dimensionless temperature increases after stabilization. For example, considering $R = 0$, the dimensionless temperatures after stabilization are 0.04, 0.07, and 0.09. Compared with the VRT, the surrounding rock temperature decreased by 98%, 96.5%, and 95.5%, respectively. Additionally, the figure indicates that the influence of thermal conductivity on the temperature field of the surrounding rock gradually diminishes with an increase in dimensionless radius. As an illustration, considering $Fo = 2$, with an increase in thermal conductivity, the dimensionless temperatures corresponding to different dimensionless radii decrease as follows: $R = 0$: 95%, 90%, 83%; $R = 1$: 33%, 30%, 27%; $R = 2$: 11%, 10%, 9%; $R = 3$: 2%, 2%, 2%, respectively.

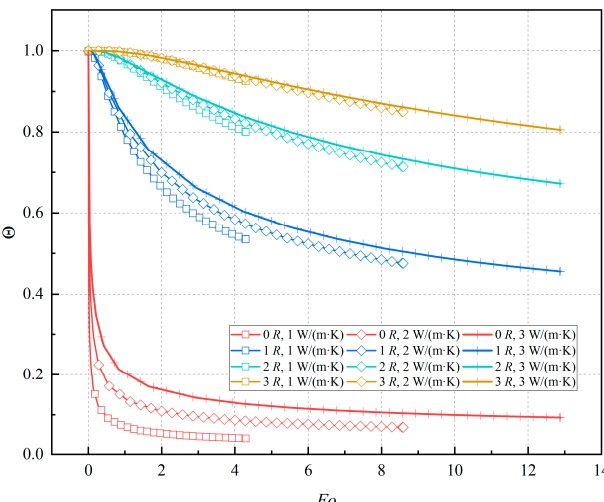

**Figure 6.** The relationship between dimensionless temperature and dimensionless time under different thermal conductivity.

### 3.3. Time–Space Coupling Characteristics of Temperature Field

Figure 7 displays the characteristic curve of the dimensionless temperature and dimensionless radius of the roadway-surrounding rock under specified conditions: air temperature of 25 °C, VRT of 50 °C, and thermal conductivity of 2 W/(m·K). The point data are the numerical simulation calculation results, and the line data are the coupling function fitting results. Given the application range of the coupling function as $R > 0$, a position near the wall of the surrounding rock is selected for study, that is, $R = 0.1$. It is observed that the fitting effect of the space–time coupling characteristic curve of the dimensionless temperature field is improved, with the fitting degree $R^2$ ranging from 0.96 to 0.99. As ventilation time increases, the fitting degree of the characteristic curve gradually improves. This occurs because, as the surrounding rock extends radially to the deep, the temperature field is gradually affected by disturbances. Especially in the early stages of ventilation, the internal temperature field of the surrounding rock remains nearly unchanged, resulting in a less obvious correlation between the data and a poorer fitting effect. At a ventilation time of 1 day ($Fo = 0.024$), the absolute error at $R = 0.1$ is $-0.028$. Simultaneously, it is evident that the dimensionless temperature gradually increases with an increase in the dimensionless radius, but the temperature rise rate gradually decreases.

Additionally, with an increase in dimensionless time, the characteristic curve shifts downward as a whole. This is attributed to the expanding disturbance range of the surrounding rock temperature field with increased ventilation time. However, the disturbance gradually weakens the temperature field of the surrounding rock due to the gradual decrease in the temperature difference between the airflow and the roadway wall. This demonstrates a gradual reduction in the downward rate of the characteristic curve.

Figure 8 presents the characteristic curve of the dimensionless temperature and dimensionless time of the roadway-surrounding rock under specified conditions: air temperature of 25 °C, VRT of 50 °C, and thermal conductivity of 2 W/(m·K). The point data are the numerical simulation calculation results, and the line data are the coupling function fitting results. The fitting effectiveness of the time–space coupling characteristic curve for the dimensionless temperature field is relatively satisfactory, with the fitting degree $R^2$ ranging between 0.95 and 0.99. With the increase in the dimensionless radius, the fitting degree of the characteristic curve decreases in the deep part. This is because the surrounding rock temperature in the deep part drops slightly, and the changing trend of the overall curve is not as obvious as that in the shallow part of the surrounding rock, resulting in a decrease in the fitting degree of the same function here. Simultaneously, the dimensionless temperature gradually decreases with an increase in dimensionless time, and the temperature drop rate also gradually decreases. For instance, the difference in dimensionless temperature

between 1 day ($Fo = 0.024$) of ventilation at $R = 0.1$ and 361 days ($Fo = 8.592$) of ventilation is 0.70, whereas the difference in dimensionless temperature between 1 day ($Fo = 0.024$) of ventilation at $R = 3.0$ and 361 days ($Fo = 8.592$) of ventilation is 0.14.

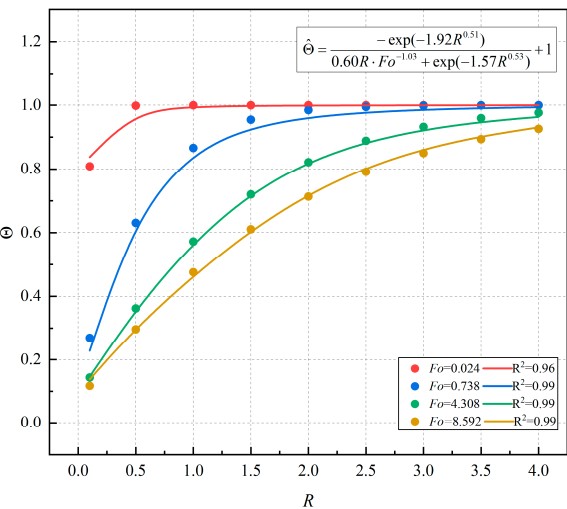

**Figure 7.** The characteristic curves of dimensionless temperature and dimensionless radius at different depths.

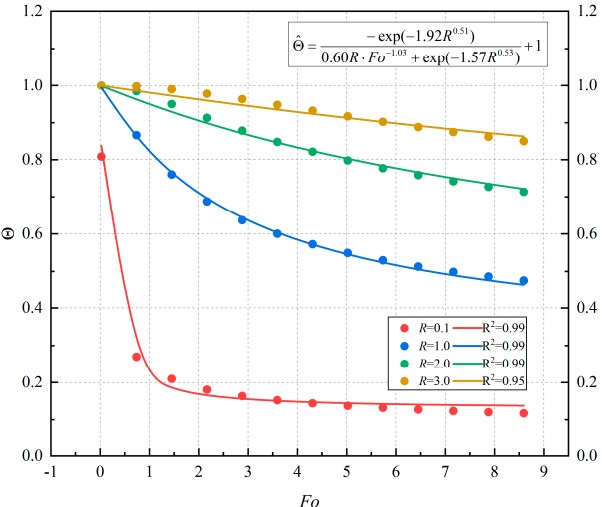

**Figure 8.** The characteristic curves of dimensionless temperature and dimensionless time at different depths.

Additionally, with an increase in dimensionless radius, the characteristic curve rises as a whole. However, as it extends to the deep part of the surrounding rock, the temperature difference between the rocks gradually diminishes, and a gradual decrease in the upward movement speed of the characteristic curve is discernible. For instance, after 31 days of ventilation ($Fo = 0.738$), the dimensionless temperature difference between the characteristic curves at $R = 0.1$ and $R = 1$ is 0.63, whereas the dimensionless temperature difference between the characteristic curves at $R = 2$ and $R = 3$ is 0.02. This occurs because, with the deep extension of the surrounding rock, the disturbance of the wind flow to the temperature of the surrounding rock gradually weakens, resulting in less noticeable temperature changes in the deep rock mass.

## 4. Discussion: The Influence of Thermal Conductivity on the Space–Time Coupling Characteristics of Dimensionless Temperature Field of Roadway-Surrounding Rock

Figure 9 illustrates the impact of thermal conductivity on the time–space coupling characteristics of the dimensionless temperature field of the roadway-surrounding rock at various time points. It is observed that the fitting degree of the characteristic curve varies between 0.90 and 0.99, and the change pattern of the characteristic curve is mainly consistent with the observations above. The fitting effect of the characteristic curve is gradually improved with increasing thermal conductivity, particularly in the early stages of ventilation. For instance, at a ventilation time of 1 day, the curve fitting performs worst with a thermal conductivity of 1 W/(m·K), and the $R^2$ is 0.90. In contrast, the curve fitting is more effective with thermal conductivities of 2 W/(m·K) and 3 W/(m·K), resulting in an $R^2$ of 0.96 and 0.98. Simultaneously, in the early stages of ventilation, fitting errors are primarily concentrated in the shallow parts. For example, at a ventilation time of 1 day, under different thermal conductivity conditions, the maximum absolute errors occur at $R = 0.1$: $-0.043$, $-0.029$, $-0.018$, and at $R = 0.5$: $0.009$, $0.017$, $0.019$. Additionally, with an increase in ventilation time, the fitting effect of the characteristic curve is gradually improved. However, the fitting error curve gradually shifts towards the depths of the surrounding rock. For instance, at a ventilation time of 181 days, under different thermal conductivity conditions, the maximum absolute error occurs at $R = 4$: $0.008$, $0.013$, $0.011$. Finally, when the ventilation days were increased to 361 days, the fitting errors were evenly distributed across locations and not concentrated in one location.

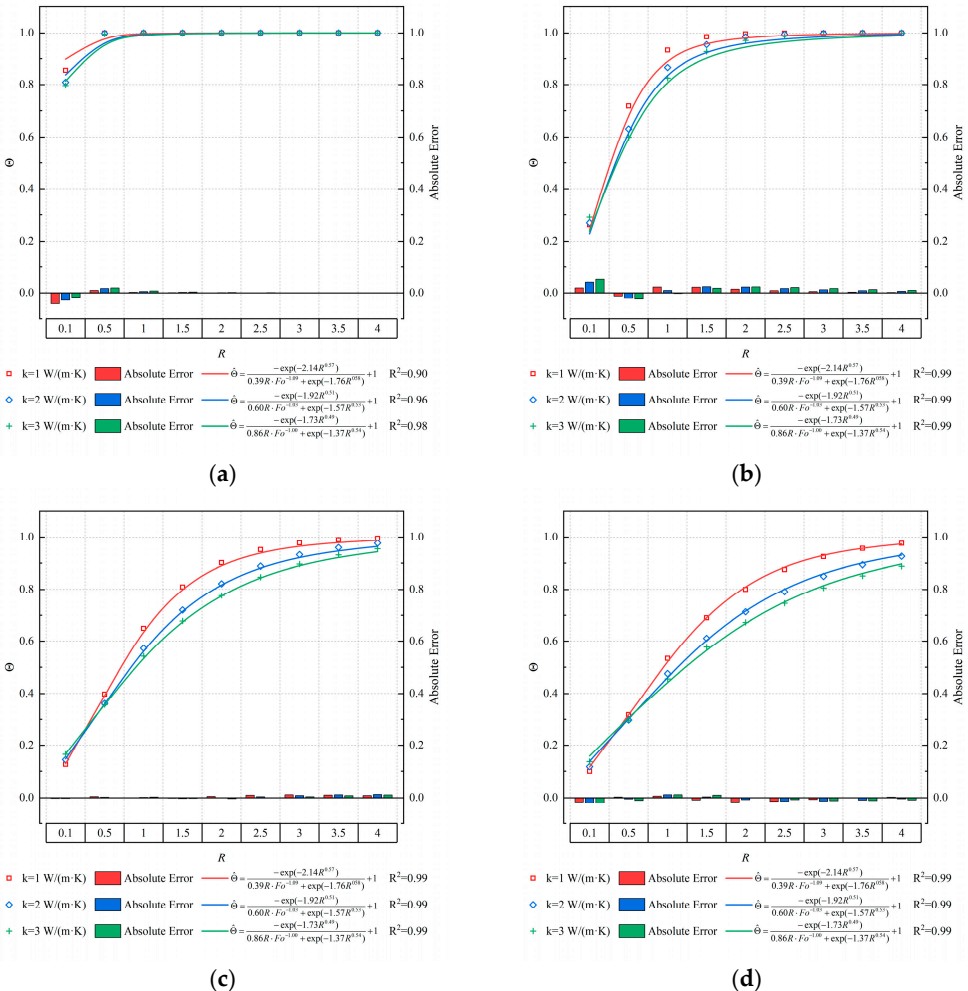

**Figure 9.** The characteristic curves of dimensionless temperature and dimensionless radius at different thermal conductivities. (**a**) 1 day; (**b**) 31 days; (**c**) 181 days; (**d**) 361 days.

Figure 10 illustrates the impact of thermal conductivity at different positions on the spatial–temporal coupling characteristics of the dimensionless temperature field of the roadway-surrounding rock. Owing to varying thermal conductivity, the corresponding *Fo* also differs at the same time. It is evident that the fitting degree of the characteristic curve gradually increases with the augmentation of thermal conductivity. For instance, at *R* = 3, with the augmentation of thermal conductivity, the fitting degree of the characteristic curve is 0.89, 0.95, and 0.97, respectively. But there are exceptions; for example, at *R* = 0.1, with the augmentation of thermal conductivity, the fitting degree of the characteristic curve is 0.99, 0.99, and 0.98, respectively.

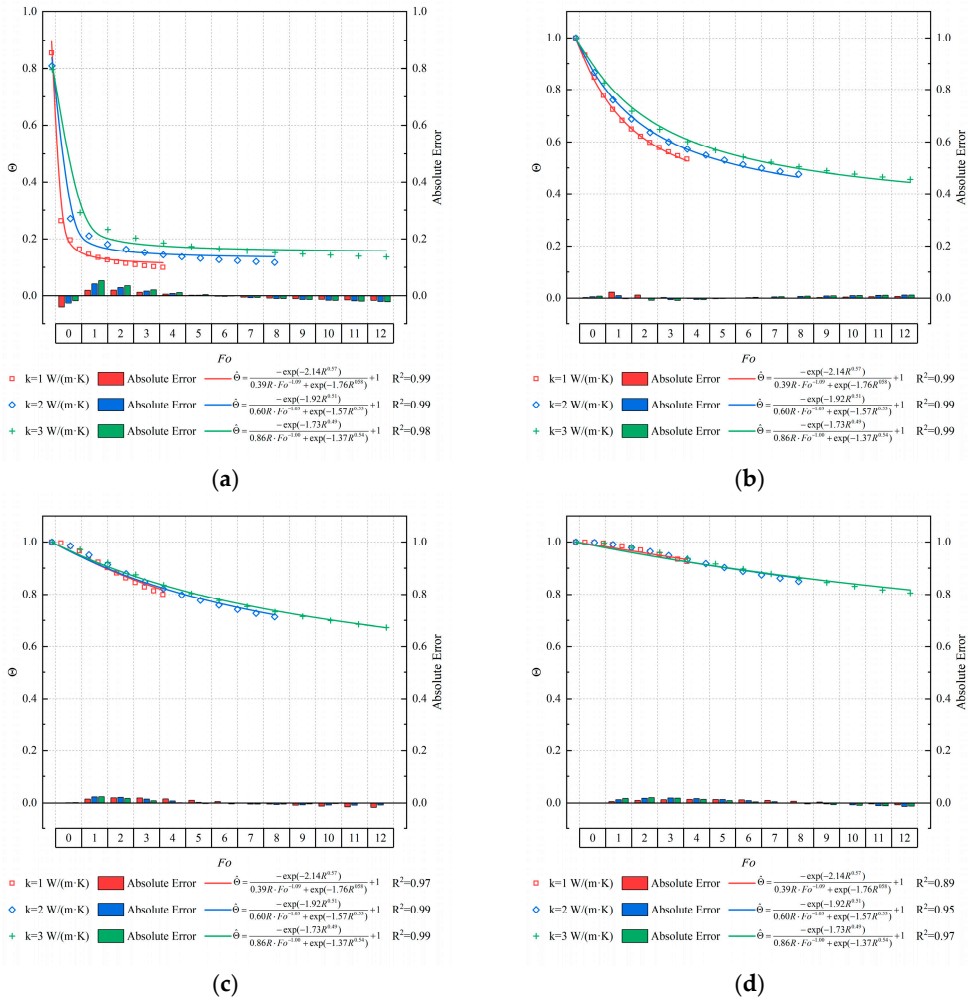

**Figure 10.** The characteristic curves of dimensionless temperature and dimensionless time at different thermal conductivity. (**a**) 0.1 *R*; (**b**) 1 *R*; (**c**) 2 *R*; (**d**) 3 *R*.

Additionally, the absolute error of the characteristic curve is observed to vary by ±0.1. Moreover, as the dimensionless radius increases, the absolute error of the characteristic curve gradually diminishes. For example, considering ventilation for 31 days with a thermal conductivity of 3 W/(m·K), as the dimensionless radius increases from *R* = 0.1 to *R* = 3, the absolute errors of the characteristic curves are 0.042, 0.001, 0.023, and 0.012, respectively. Moreover, with an increase in dimensionless time, the absolute error of the characteristic curve initially increases and then decreases, with the maximum absolute error primarily occurring during ventilation from 31 days to 91 days.

In conclusion, the fitting equation $\Theta = \dfrac{-\exp\left(A * R^B\right)}{O * R * Fo^P + \exp\left(S * R^Q\right)} + 1$ demonstrates a strong fitting effect regarding the spatial and temporal distribution of the surrounding rock temperature field under different thermal conductivity conditions. Furthermore, with an increase

in thermal conductivity, the fitting degree of the formula improves. Consequently, the equation can effectively describe the time–space coupling characteristics of the temperature field of the roadway-surrounding rock.

## 5. Conclusions

In this study, a two-dimensional heat transfer model of the roadway-surrounding rock is formulated, and the temporal and spatial evolution of the temperature field of the roadway-surrounding rock is scrutinized. Utilizing the Gauss–Newton method, the temporal and spatial coupling characteristics of the temperature field of the roadway-surrounding rock and the impact of thermal conductivity on it are examined. The key conclusions are outlined below:

(1) The evolution law of the temperature field of the roadway-surrounding rock in an unventilated state is examined. The findings reveal that in the spatial dimension, the dimensionless temperature of the surrounding rock gradually increases, and the rate of temperature rise diminishes until it reaches the initial temperature of the original rock during the radial extension of the surrounding rock into the deep. In the time dimension, as the ventilation time increases, the infinite temperature of the surrounding rock gradually decreases, and the cooling rate progressively diminishes until it stabilizes at a fixed value.

(2) The space–time coupling characteristics of the temperature field of the roadway-surrounding rock are examined. The findings demonstrate that the function $\Theta = \frac{-\exp\left(A \times R^{B}\right)}{O \times R \times Fo^{P} + \exp\left(S \times R^{Q}\right)} + 1$ can represent the coupling characteristics of the surrounding rock temperature field. In the time dimension, as the ventilation time increases, the fitting degree of the characteristic curve gradually improves. The characteristic curve moves downward as a whole, leading to a gradual decrease in dimensionless temperature and a reduction in the temperature drop rate. In the spatial dimension, with the increase in the dimensionless radius, the fitting degree of the characteristic curve decreases in the deep part. The characteristic curve moves upward as a whole, resulting in a gradual increase in dimensionless temperature while the temperature rise rate gradually decreases.

(3) This study investigates the impact of thermal conductivity on the temporal and spatial coupling characteristics of the temperature field in the surrounding rock of roadways. The findings indicate that a higher thermal conductivity corresponds to a higher fitting degree of the coupling characteristic function of the surrounding rock. Moreover, under the same ventilation time, larger thermal conductivity results in a smaller wall cooling range, a greater internal temperature drop rate of the surrounding rock, and a larger range of disturbed surrounding rock.

**Author Contributions:** Conceptualization, J.Z. and Y.Z.; methodology, J.Z.; validation, P.S. and Y.L.; investigation, Y.L.; writing—original draft preparation, J.Z.; writing—review and editing, Y.Z.; visualization, P.S.; funding acquisition, Y.Z. All authors have read and agreed to the published version of the manuscript.

**Funding:** This research was funded by the National Natural Science Foundation of China (52074266).

**Institutional Review Board Statement:** Not applicable.

**Informed Consent Statement:** Not applicable.

**Data Availability Statement:** The original contributions presented in the study are included in the article, further inquiries can be directed to the corresponding author.

**Conflicts of Interest:** The authors declare that they have no conflicts of interest.

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
