# Peer review of "Dimensionless Analysis of the Spatial–Temporal Coupling Characteristics of the Surrounding Rock Temperature Field in High Geothermal Roadway Realized by Gauss–Newton Iteration Method"

_applsci, doi:10.3390/app14041608_

Round 1
Reviewer 1 Report
Comments and Suggestions for Authors
The concept of the research is important in mines at high-temperature rocks, however, its importance is not sufficiently detailed. It seems that the title has some missing word(s) why we have evolution, what has surrounding rock, and what you mean by "high geothermal".
(1) The state-of-art is not related directly to the research. If you write a general temperature field analysis research, the state-of-art must contain more methodological analysis instead of the real results. Please use paragraphs in the state-of-art.
The methodology part is inadequate.
(2) The main calculation method is missing, we don't know whether it is based on analytical or numerical calculation, it is based on personal calculation or software calculation.
(3) In Chapter 2.1., the figure is not proportional, the table contains thermal diffusivity values which are not used later and the relation between the thermal conductivity and diffusivity values is not clear.
(4) In Chapter 2.2., there are missing descriptions of letters (like T_f, tau), a is thermal diffusivity instead of thermal diffusion coefficient (everywhere in the text), how can you define the radius of a rectangle?
The results are too general and not contain new information (presumably the pure results of the not-presented calculation, which can be found in heat conduction books).
(5) Results are one-dimensional.
(6) Based on the Chapter 2.2., If R<1, the position is inside the tunnel, so the temperature function is different than in the rocks, but there is no sign of it on the images
(7) Images suggest fixed temperature boundary condition instead of adiabatic (which is mentioned in the methods.)
(8) In chapter 3.3., I am not sure that Eq (7) and (8) come from the cited sources and not clear the derivation of Eq (9) from them. In addition, this is a methodology, so it has to move to Chapter 2.
(9) What is the origin of the Fig. 6?
(10) What is the origin of the dotted data in Fig. 7 and 8?
(11) The result of the iteration is not detailed except in Fig. 7 and 8. I can't accept this function as a good approximation because it suggests, that because of the cooling effect of the tunnel, the temperature of the surrounding rock could be higher than the original temperature.
Without a detailed methodology, the research is not reproducible. The aim of the result is not clear, why it is better than the previous one. The results need some comparison with case studies or other results with different calculations (it is suggested in the text as well, but missing).
Based on the above I suggest describing a more specific aim for the research, giving a more accurate methodology, and based on them please make a detailed discussion which presents the relevance of the research/results.
Comments on the Quality of English LanguageSeveral sentences are too long and complex, please make a complex revision on the content.
Author Response
Thank you for your review comments on this paper.
Please see the attachment

Reviewer 2 Report
Comments and Suggestions for Authors
You should avoid writing equations in the Abstract.
Line 133 “Disregard the contact thermal resistance between the model's layers.” - Isn't this an oversimplification?
Line 141, Table 1 - I would like to ask for a more detailed explanation of the term "unstable heat transfer coefficient".
Table 1 – “Thermal diffusion coefficient of surrounding rock” or thermal diffusivity coefficient?
Line 157 - Thermal diffusivity coefficient should be not “the thermal diffusion coefficient”.
Line 213 - At this point, a Figure would be useful to illustrate this description: "At 361 days of ventilation, as the dimensionless radius increases from R=1 to R=4, the temperature of the surrounding rock decreases by 95%, 55%, 70% , and 18%, respectively, compared to the virgin temperature." Unfortunately, this is not clear from Figure 3.
Line 275 – “Where C, D, F, F, G, H are constants.” – Is The “F” different from another “F”?
Line 310 – Figure 7 – This figure lacks any match for these two positions – “1 day”, and “31 day”. This must be corrected.
Line 316 – R2 – equals 0.82 – is definitely not satisfactory. As you can see in Figure 8.
Line 347 – Accuracy R2 – in range 0.82-0.83 is very poor, you know it.
Figure 9 is unreadable, but as can you see the “1 day” and “181 day” are totally poorly matched.
Figure 10- is unreadable, but as you can see in the “181 day” there is a completely different course of points than matches, this cannot be the case.
The work is quite interesting, but the authors should work more on accuracy, and as you can see, there is a very big problem with this work. The work is purely theoretical, so I believe it should be thoroughly revised and corrected. Unfortunately, in my opinion, it is not suitable for publication in this form.
Author Response
Thank you for your review comments on this paper.
Please see the attachment.

Reviewer 3 Report
Comments and Suggestions for Authors
In this paper, the authors investigated the employing of the Gauss-Newton iteration method to model the coupling relationship between temperature, time, and space.
They obtained an expression of the temperature that effectively describes the temperature field changes in both time and space dimensions.
The authors found that, over time, as ventilation duration increases, the fitting degree of the characteristic curve steadily rises, and the characteristic curve to descend overall. In the spatial dimension, the fitting degree of the characteristic curve gradually increases with the rise of the dimensionless radius, and the characteristic curve to ascend overall. Additionally, as thermal conductivity increases, the fitting degree of the characteristic curve steadily rises.
The results obtained in this paper are useful to understanding the time-space coupling characteristics of the surrounding rock temperature field in high geothermal roadways, therefore these results could have an important role for controlling heat damage in mines.
Observations:
1). Page 4, line 156. Instead of ‘f0” must be “F0”.
2). Verify and correct Eq.(6). The parameter “miu_l” must be multiplied by the Unit matrix “I” of order “n”.
3). As the authors claim on page 4, line 161, the dimensionless temperature “Theta” is a function of two variables, namely the non-dimensional time “F0” and the non-dimensional spatial variable “R”. Such problem is called an one-dimensional problem, because the unknown function depends only of one spatial variable. It is not a two-dimensional problem as the authors wrote on page 13, line 398.
Author Response

(The authors gave the same response as above.)

Round 2
Reviewer 1 Report
Comments and Suggestions for Authors
Thank you for accepting my opinion about improving your manuscript. However, there are some problems wich are not handled properly.
(1) The problem of the title remains: too general and the last word is an adjective, without a noun
(2) In Line 133: please cite reports.
(3) In Chapter 2.2.: Bi is Biot number instead of Biwell. Why the eq. (1) can be used in this geometry (the self-reference is not enough, because the system is different and there is no verification in it). Check that kinematic viscosity is not "v" but greek "nu".
(4) Based on the previously given data, thermal diffusivity of the 1st program is 0.66E-6, instead of 0.33E-6.
(5) If the R is calculated with the presented way, R=0 is inside the tunnel, not in the wall, but that is not appear in the images (the problem remains).
(6) What is the origin of the values presented in Fig. 2.? If it comes from this research, it could be placed in the result.
(7) In line 229, change "°C" to W/(mK)
(8) In line 238&239 the number of eq.s is wrong.
(9) What is the position of the points in the Fig. 3. and why? Is it different than the "observation points"?
(10) In the case of Fig 3 & 4 there is a mismatch between data. E.g. for R=2 and Fo=4.3 theta is 8 in Fig. 3 and 0.58 in Fig 4. Why?
(11) In Fig 7 I try to use the given function for the measured values, and I got a significantly different shape of curves. E.g. In the case of R=1 and Fo=4,308, the result of the function is around 0.87, trhe value of the curve 0.56, while the datapoint is around 0.59. (And the value for these parameters are 0.8 in Fig 3 and 0.1 in Fig. 4)
(12) What are the R^2 values in your interpretation? They are too high, but the curves are not the same than the dots.
(13) In some images the captions of fuctions are too small, we can't see or check them.
(14) What is the relation of the results for the other researches?
(15) What is the practical usage of the results? They are too general and there is no applied part.
(16) How does the research related to "New advanced in mining technology"?
Database is not clear and not coherent based on the images.
Comments on the Quality of English LanguagePlease use space before [.
Line 129 researches instead of research.
Author Response
请参阅附件。

Reviewer 2 Report
Comments and Suggestions for Authors
The manuscript has been corrected, thank you.
Now it presents a higher level of advancement.
The model looks well-fitted now.
I believe that the presented Manuscript is now suitable for publication.
Author Response
Thank you for your review comments on this paper.
Your expertise and insightful comments have greatly contributed to enhancing the quality of my work.